# The Regulation of MicroRNA-21 by Interleukin-6 and Its Role in the Development of Fibrosis in Endometriotic Lesions

**DOI:** 10.3390/ijms25168994

**Published:** 2024-08-19

**Authors:** Maria Ariadna Ochoa Bernal, Yong Song, Niraj Joshi, Gregory W. Burns, Emmanuel N. Paul, Erin Vegter, Samantha Hrbek, Lorenzo F. Sempere, Asgerally T. Fazleabas

**Affiliations:** 1Department of Obstetrics, Gynecology & Reproductive Biology, Michigan State University, Grand Rapids, MI 49503, USA; ochoaber@msu.edu (M.A.O.B.); songyon3@msu.edu (Y.S.); joshini@msu.edu (N.J.); burnsgr2@msu.edu (G.W.B.); paulemma@msu.edu (E.N.P.); vegterer@msu.edu (E.V.); bondsama@msu.edu (S.H.); 2Department of Animal Science, Michigan State University, East Lansing, MI 48824, USA; 3Precision Health Program and Department of Radiology Michigan State University, East Lansing, MI 48824, USA; semperel@msu.edu

**Keywords:** endometriosis, microRNA-21, IL-6, inflammation, fibrosis

## Abstract

Endometriosis is one of the most common causes of chronic pelvic pain and infertility that affects 10% of women of reproductive age. It is currently defined as the presence of endometrial epithelial and stromal cells at ectopic sites; however, advances in endometriosis research have some authors believing that endometriosis should be re-defined as “a fibrotic condition in which endometrial stroma and epithelium can be identified”. microRNAs (miRNAs) are regulatory molecules that potentially play a role in endometriotic lesion development. There is evidence that suggests that miRNAs, including microRNA-21 (miR-21), participate in fibrotic processes in different organs, including the heart, kidney, liver and lungs. The objective of this study was to understand the role of miR-21 and the mechanisms that can contribute to the development of fibrosis by determining how IL-6 regulates miR-21 expression and how this miRNA regulates the transforming growth factor beta (*TGF-β*) signaling pathway to promote fibrosis. We investigated the expression of miR-21 in the baboon and mouse model of endometriosis and its correlation with fibrosis. We demonstrated that inflammation and fibrosis are present at a very early stage of endometriosis and that the inflammatory environment in the peritoneal cavity, which includes interleukin 6 (IL-6), can regulate the expression of miR-21 in vitro and in vivo.

## 1. Introduction

Endometriosis is an inflammatory, fibrotic and estrogen-dependent gynecological disorder characterized by endometrial-like tissue outside the uterus, mainly on the pelvic peritoneum, ovaries and rectovaginal septum. In rare cases, this endometrial-like tissue can be found in uncommon areas such as the pericardium, pleura, diaphragm or even the brain [1,2,3,4]. Endometriosis affects approximately 5–10% of women of reproductive age and is associated with pelvic pain and infertility [2,5]. Endometriosis has a varying degree of symptoms and is challenging to diagnose; it can take up to 7–10 years before the diagnosis [6], and the symptoms can vary widely. For some women, it is probable that endometriosis begins during their adolescent years and is diagnosed later in their adult life [7]. Women may be asymptomatic or present with a single or a combination of symptoms with different intensities that can easily be attributed to other conditions [5]. This is one of the factors that contribute to the delay of the diagnosis. One of the symptoms that is associated with endometriosis is painful menstruation (dysmenorrhea), which is widely experienced in adolescents affected by endometriosis. Some studies have suggested that dysmenorrhea at the early stages of adolescence could be associated with the subsequent development of endometriosis [8]. Other symptoms also associated with endometriosis are cyclical or non-cyclical abdominal pain, recurrent painful urination (dysuria), pain during and after sexual intercourse (dyspareunia), painful defecation (dyschezia), gastrointestinal discomfort and/or decreased libido [9]. Currently, there are no reliable biomarkers available to diagnose this disease. The peritoneal fluid in women with endometriosis contains abundant proinflammatory cytokines, particularly interleukin 6 (IL-6), which creates a chronic inflammatory state [10]. IL-6 is a soluble mediator with a pleiotropic effect on inflammation and immune response [11]. The activation of its receptor (IL-6R) triggers the activation of downstream signaling molecules, including signal transducer and activator of the transcription 3 *(STAT3)* [11,12,13]. *STAT3* is located in the cytoplasm until it is activated by phosphorylation. After activation, it translocates to the nucleus and binds to promoter regions for target gene expression [14]. *STAT3* is an important key transducer and regulator of genes that play important roles in processes such as cell development, proliferation, survival and inflammatory processes [15]. The role of *STAT3* during the development of endometriosis is unclear, but studies have shown high levels of phosphorylated *STAT3 (pSTAT3)* in the endometrium of women with endometriosis compared to controls [14]. Other studies in baboons and mice have also shown that the phosphorylation of *STAT3* can be increased by IL-6 [14,16,17,18].

In addition, other studies have suggested that microRNAs (miRNAs) play a role in non-malignant and malignant diseases that involve the human reproductive tract. Abnormal expression of miRNAs has been observed in several reproductive diseases, including preeclampsia, endometrioid endometrial adenocarcinoma and endometriosis [19,20,21]. Studies on miRNA expression support the hypothesis that miRNAs play an important role during the progression of endometriosis [19]. miRNAs regulate processes such as progesterone resistance, inflammation, cell proliferation, extracellular matrix remodeling and angiogenesis, amongst others that are key for the development of endometriosis [22].

MicroRNA-21 (miR-21) plays an important role in different biological functions, including cell proliferation, fibrosis, migration, apoptosis, invasion and inflammation [23]. All these processes, in which miR-21 intervenes, play an important role in the development and progression of endometriosis. However, the involvement of miR-21 in this disease is still unclear.

MiR-21 is one of the miRNAs that was found to be upregulated in different fibrotic diseases and is associated with TGF- β signaling [24]. Studies have shown a positive association between miR-21 and connective tissue growth factor (CTGF), emphasizing the role of miR-21 during the development of fibrosis [25].

However, it is unknown if it plays a role in the fibrogenic process associated with endometriosis. In this study, we propose that dysregulation of miR-21, driven by the upregulation of IL-6 during the development of endometriosis and consequent activation of *STAT3*, can modulate TGF-β signaling in endometriotic lesions by amplifying components of TGF-β signaling resulting in the development of fibrosis.

## 2. Results

### 2.1. Progression of Endometriosis in the Mouse Model

To understand the progression of endometriosis and the changes over time, we induced endometriosis in 6–8-week-old female mice. Endometriotic lesions were collected during diestrus at three different time points (15 days, 1 month and 3 months) after the induction of endometriosis. Using the double-fluorescence cell tracer Cre/LoxP systems, Pgr cre/+ Rosa26 mT/mG, lesion sites could be clearly visualized during the development of the disease, and the number of lesions was counted at each time point (Figure 1A). Most lesions were observed within the peritoneal cavity. We observed that the number of lesions increased during the development of endometriosis, showing a significant difference between one month and three months (Figure 1B). No differences were observed in lesion weight at the three different time points (Figure 1C).

### 2.2. Development of Fibrosis in the Mouse and Baboon Models

To gain insight into the progression of fibrosis over time in the mouse and baboon models, we performed Masson’s trichrome staining to quantify the extent of fibrosis in endometriotic lesions. In the mouse model, Masson’s trichrome staining revealed the deposition of collagen surrounding the lesion during the three time points (15-day, 1-month and 3-month lesions) (Figure 2A,C). In order to validate our findings from the mouse model in the baboon during the course of the disease, Masson’s trichrome staining confirmed an increase in the deposition of collagen in lesions at 15 months (Figure 2B,D). These observations showed that the extent of fibrosis is progressively increased in endometriotic lesions and eutopic endometrium in baboons with endometriosis (Figure 2B).

### 2.3. Small-RNA Library and Ingenuity Pathway Analysis (IPA)

To identify miRNAs that are differentially expressed, we performed small RNA-sequencing and found that a total of 586 miRNAs were expressed (≥2 CPM, ≥2 samples) in the baboon model with 256 miRNAs (44%) differentially expressed (DE), (FDR < 0.05) between matched ectopic lesions and eutopic endometrium. When the eutopic endometrium was compared to spontaneous lesions, 233 of the 571 miRNAs were DE (41%), mirroring the differences identified in the induced model. A comparison of lesions from spontaneous (*n* = 8) versus induced endometriosis (*n* = 16) found 537 miRNAs with only 10 DE miRNAs (1.9%).

Based on the similarity of the induced and spontaneous disease, all samples were combined (*n* = 24), and we found that 304 of 579 miRNAs were DE (52%), with 150 increased in the ectopic lesions compared to the eutopic endometrium. Among the top DE miRNAs related to fibrosis, miR-21 was identified. The Ingenuity Pathway Analysis of the 304 DE miRNAs identified angiogenesis and organismal injury, including fibrosis, as the major functional alterations in the predicted gene networks. Ingenuity Pathway Analysis of DE miRNAs identified enrichment for associations with endometriosis (*p* = 2.1 × 10^−10^) and activation of fibrosis (*p* = 1.2 × 10^−9^) (Figure 3).

### 2.4. miR-21 Is Increased in Endometriotic Lesions

To characterize the expression of miR-21 in vivo in the mouse model, we first performed RT-qPCR on matched samples, uterus (control eutopic)–ectopic (lesion) and eutopic (post-inoculation uterus)–ectopic (lesion) at the different time points of collection (15 days *n* = 7, 1 month *n* = 10 and 3 months *n* = 9). The RT-qPCR analysis revealed that the expression of miR-21 was increased in lesions compared with the matched uteri of the mice and increased over time up to 3 months lesions (Figure 4A).

We validated these results in baboons with induced disease (*n* = 4). We observed an increase in miR-21 in ectopic lesions compared with the eutopic endometrium at 15 months after endometriosis induction (Figure 4B), but this increase was not significant.

We also analyzed matched mid-secretory eutopic endometrium and ectopic lesions from women with deep infiltrating endometriosis (DIE). We observed an increasing trend in the expression of miR-21, but this increase was not significant (Figure 4C). Our data suggested that miR-21 is increased in mouse, baboon and human lesions, but was only statistically significant in the mouse.

### 2.5. miR-21 Is Predominantly Expressed in the Stroma of Endometriotic Lesions

To identify the cell-specific location of miR-21 in vivo, we next performed in situ hybridization on 3-month lesions from mice (Figure 5 and Figure 6) and matched mid-secretory eutopic and ectopic tissues in the baboons at 15 months after endometriosis induction (Figure 7). In the mouse model, the in situ hybridization analysis revealed that miR-21 was predominantly expressed in the stromal cells. (Figure 5). This analysis was performed using cell segmentation analysis. The tissue was segmented based on DAPI counterstaining. miR-21 expressing cells were displayed in blue, yellow or red depending on their low to high-intensity staining. We also observed a higher expression of miR-21 in the lesions compared to the control uterus. Stromal cells showed a higher intensity of the miR-21 expression in the lesion compared to the uterus. 18SrRNA was used as a reference noncoding RNA control for the uterus and the lesions.

### 2.6. miR-21 Expression and Fibrosis Development in the Mouse Model

In order to determine the role of miR-21 during the development of fibrosis within endometriotic lesions, we used the in situ hybridization analysis of the 3 months mouse lesions (Figure 6A) and compared it with Masson’s trichrome staining of the same lesions (Figure 6B). In the lesions, we observed that the areas where collagen was accumulated (represented in blue) with trichrome staining corresponded with areas where miR-21 was present by in situ hybridization, suggesting an association between the expression of miR-21 and the presence of collagen within the lesions.

### 2.7. miR-21 Expression in the Baboon Model

In situ hybridization in the baboon (Figure 7A, left panel) shows the expression of miR-21, cytokeratin and U6 in the eutopic endometrium of the baboon, and the right panel shows the ectopic lesion from the baboon. miR-21 is represented in green, cytokeratin in pink and U6 in yellow. The two bottom pictures in Figure 7A represent the co-detection of miR-21, the reference noncoding RNA U6 and cytokeratin. In these micrographs, the DAPI signal was used as nuclear counterstaining. The expression of miR-21 in the endometriotic lesions is higher than in the eutopic tissue, being significant in the stromal region (Figure 7B, (*p* < 0.05)).

### 2.8. CTGF Expression Is Correlated with the Presence of Fibrosis in the Mouse Model, Baboon Model and Women with Endometriosis

*CTGF* plays an important role during the development of fibrosis. In order to explore the characterization of *CTGF* in vivo, we performed RT-qPCR on matched samples obtained at 15 days in mice, matched mid-secretory eutopic endometrium and ectopic lesions from baboons and women with endometriosis. The RT-qPCR analysis revealed that the expression of *CTGF* was significantly increased in the mouse model (*p* < 0.05) (Figure 8A), baboon (*p* < 0.001) (Figure 8B) and human (*p* < 0.001) (Figure 8C) in endometriotic lesions compared to the eutopic endometrium.

### 2.9. IL-6 Upregulates miR-21 in the Mouse Model

Following the induction of endometriosis in the mouse model (Figure 14A), we treated the animals with IL-6 (5 μg/injection) or PBS in the peritoneal cavity for 15 days (Figure 14D). We observed that the mice treated with IL-6 had a significant increase in the number of lesions compared to the PBS controls (Figure 9A,B). The weight of the lesions pooled together was also higher in response to IL-6 treatment (Figure 9C). In addition, the expression levels of miR-21 in the animals injected with IL-6 were significantly higher than in the animals injected with PBS (Figure 10A). The lesions were easily visualized in green under the fluorescence microscope (Figure 10B).

### 2.10. IL-6 Upregulates miR-21 via p-STAT3 in Ectopic Stromal Cells

To confirm that interleukin-6 (IL-6) can activate STAT3 and subsequent phosphorylation in our cell lines, we first initiated a dose–response with 25 ng/mL and 50 ng/mL of IL-6 (Figure 11A). Both concentrations resulted in a positive response, but 50 ng/mL had a higher response. A time course experiment (12, 24 and 48 h) with 50 ng/mL was also performed. Following IL-6 stimulation at 50 ng/mL, miR-21 was initially increased (Figure 11B) at 12 h, and one of its targets, *Smad 7*, was significantly decreased.

We also investigated the effect of IL-6 in vivo using the mouse model of endometriosis. In this case, not only was miR-21 increased in lesions following 2 weeks of IL-6 treatment, but this was also accompanied by an increase in the number of lesions and an increase in the total weight of lesions. The fluorescence reporter genes that the mouse model utilized in this study allowed us to visualize endometrial lesions in vivo. Both in vitro and in vivo studies showed similar responses confirming the role of IL-6 in regulating the expression of miR-21.

To confirm these results in silico, *STAT3* binding sites on miR-21 promoter regions (Figure 12A) on chromosome 17 were identified using the genome browser Ensembl and the transcription factor binding profile database JASPAR CORE 2022. To confirm this in silico observation, ectopic stromal cells (~100.000 cells) were processed for CUT&RUN assay by treating the cells with IL-6 and using *p-STAT3* antibody.

Ectopic stromal cells were treated with IL-6 (50 ng/mL) for 12 h and were harvested for the CUT&RUN assay. This in vitro experiment confirmed the regulation of miR-21 by IL-6 via *p-STAT3* in the ectopic stromal cells (Figure 12B). From the motif map analysis, we found predicted *STAT3* binding sites on the human miR-21 promoter region (Figure 12C,D). The CUT&RUN assay confirmed that *STAT3* binds the transcription factor sites and revealed a 30-fold enrichment for *p-STAT3* compared with the IgG control antibody (Figure 12E).

### 2.11. Upregulation of miR-21 in Ectopic Stromal Cells

Our in vitro data from the treatment of ectopic stromal cells with IL-6 demonstrated an increase in miR-21 expression and a decrease in *Smad7* at the mRNA level. To verify this, we transfected stromal cells with the miR-21 mimic or nontargeting negative controls. The transfection with miR-21 mimic resulted in an increased detection of miR-21 (Figure 13A) and a decrease in *SMAD7* mRNA (Figure 13B), which confirmed the results obtained with the IL-6 treatment.

## 3. Discussion

Our study showed that miR-21 was upregulated in the stroma of endometriotic lesions in mice. In addition, we observed an association between miR-21 and fibrosis. Previous studies have shown a clear relationship between miR-21 and fibrosis in different diseases, such as cardiac fibrosis [26,27], scleroderma fibrosis [28] and lung fibrosis [29], amongst others. This study is the first to show a correlation between miR-21 and the development of fibrosis during the progression of endometriosis.

Fibrosis is one of the established hallmarks of endometriosis [30,31]. The development of fibrosis in endometriotic lesions is a complex phenomenon with underlying mechanisms that still remain to be understood [3]. Fibrosis is present in different types of endometriosis, such as peritoneal fibrosis, ovarian fibrosis, deep infiltrating endometriosis (DIE) or adhering tissue. All these different manifestations of fibrosis could lead to pain, anatomical distortion or infertility [32].

MiR-21 is a well-established pro-fibrotic miRNA [33]. Studies have demonstrated that elevated expression of miR-21 may play an important role in the development of fibrosis [34]. Previous studies have demonstrated the importance of miRNAs during the progression of endometriosis [35]. The small-RNA-seq analysis of baboon lesions performed in this study showed that a group of miRNAs are dysregulated in endometriotic lesions, including specific ones that are associated with fibrosis. One of the dysregulated miRNAs found in our analysis was miR-21, which may relate to the establishment of fibrosis in patients with endometriosis.

To study the relevance of miR-21 throughout the development of fibrosis during the progression of endometriosis, we utilized the Pgr cre/+ Rosa 26 mT/mG mouse model of endometriosis, which develops endometriotic lesions similar to the ones observed in humans [36]. Utilizing this mouse model, we observed that the number of lesions that developed during the progression of endometriosis increases as the disease advances. In addition, we also utilized this model to quantify the levels of miR-21 expression in the lesions compared with matched uteri and confirmed that it was significantly upregulated after one month of induction of endometriosis. We observed that after three months of induction, there were no differences. This could be due to the type of lesions selected for RT-qPCR. In the mouse model of endometriosis, during the development of the disease, as in women, there are different types of lesions that are evident at the time of collection. As the disease advances, the lesions become more cystic. These types of lesions contain less stroma compared to lesions that are collected at earlier time points. At three months following the induction of endometriosis, we observed several lesions that were cystic.

When we compared the miR-21 expression results in the mouse with the baboon and the human lesions, we observed an increase in the levels of miR-21 expression, although the increase in the baboon and the human was not statistically significant. Different studies have shown that miRNA expression patterns may differ depending on the endometriotic lesion type [37]. In this study, lesions collected from the mouse and baboon were primarily peritoneal lesions. In the case of the women with endometriosis, the samples utilized for this analysis were derived from women with DIE. This finding suggests once again the importance of identifying the nature of the endometriotic lesions and the heterogeneity in the population of women with endometriosis.

Our in situ hybridization results showed a high expression of miR-21 in the stromal compartment of the mouse and baboon lesions. The location of miR-21 matched the deposition of collagen based on the Masson trichrome staining, indicating an association with fibrosis. One of our parallel studies also confirmed this deposition of collagen in mouse endometriotic lesions using a gadolinium-based collagen I targeting probe [38]. Other studies have also shown an upregulation of miR-21 expression predominantly in the stromal cell compartment of tumor-associated fibroblasts [39,40]. In addition, other groups have suggested that miR-21 can influence fibrogenic processes in adenocarcinoma [41,42] and colorectal cancer [43]. These studies match our findings related to the role of miR-21 in fibrogenesis and support the hypothesis that miR-21 plays a role in the development of fibrosis in endometriosis.

Another component that was shown to be relevant in our studies was the presence of *CTGF*. The production of *CTGF*, a key marker in fibrotic disorders such as endometriosis and intrauterine adhesions, is through STAT3-dependent Smad signaling [44]. RT-qPCR data from mouse, baboon and women with endometriosis showed that *CTGF* is highly expressed in the stromal cells. *CTGF* has also been reported to enhance the mRNA expression of collagen I and fibronectin in fibroblasts [45]. It has also been reported that *TGF-β* and *CTGF* are downstream mediators of TGF-β regulated fibroblast cell growth, extracellular matrix secretion and enhancing production of collagen [46]. In addition, *TGF-β* regulates the transcription of several genes involved in fibrosis, including *CTGF* [47], and studies have suggested that *CTGF* acts as a mediator of *TGF-β*-induced fibrotic pathways via miR-21 regulation [25]. Our data support a potential connection between miR-21 and *CTGF* during the development of fibrosis in endometriosis.

Several studies have shown elevated levels of IL-6 in peritoneal fluid of patients with endometriosis [48,49], which led us to investigate whether the expression of miR-21, via *STAT3*, is induced by IL-6.

Cytokines can contribute to the pathophysiology of endometriosis in at least two ways: by enhancing the establishment and proliferation of ectopic endometrial implants and by influencing the secretion of cytokines by macrophages [50]. IL-6 is a multifunctional cytokine that stimulates cell proliferation and is involved in the formation of adhesions. IL-6 is elevated in the peritoneal fluid, endometriotic lesions and serum from women with endometriosis [51,52]. IL-6 exerts multiple bioactivities through its receptor (IL-6R). The membrane-binding receptor (mIL-6R) and the soluble receptor (sIL6R) are two forms of IL-6R. The biological activity of IL-6 is mainly mediated through binding with the corresponding mIL-6R [52].

Macrophages are the predominant cells secreting IL-6 in peritoneal fluid [53], but B cells are also implicated in its secretion [54]. The local pelvic inflammatory process, accompanied by altered function of immune-related cells and changes in cytokine content in the peritoneal cavity, has been shown to be related to the development of endometriosis [55]. There are a variety of cytokines, including IL-6, that can activate the signal transducer and activator of the transcription (STAT) family of transcription factors, such as *STAT3* [56].

*STAT3* is located in the cytoplasm until it is activated by phosphorylation. Once its activation occurs, it translocates to the nucleus and binds to promoter regions for target gene expression [57], including the expression of miR-21. In our in vitro experiments, we confirmed this by treating the cells with IL-6. We observed a higher expression of *p-STAT3* in the cells treated with IL-6, which resulted in increased expression levels of miR-21. In addition, our CUT&RUN experiments confirmed the regulation of miR-21 expression by IL-6 via *p-STAT3* in the ectopic stromal cells.

Our study has provided four main findings: First, the expression of miR-21 was increased in endometriotic lesions in the mouse and baboon compared with the eutopic endometrium, which was likely associated with the involvement of miR-21 in the process of fibrosis during the development of endometriosis. Second, *CTGF* and collagen deposition are increased during the development of the disease and are associated with the fibrotic process. Third, the localization of miR-21 in the stroma of endometriotic lesions matches with the deposition of collagen, suggesting that miR-21 is involved in the process of fibrosis during the progression of endometriosis. Finally, the activation of *STAT3* via IL-6 signaling that upregulates miR-21 leads to an increase in fibrosis. These findings highlight miR-21 as a potential diagnostic and prognostic marker and therapeutic target for fibrotic diseases such as endometriosis.

This study had some limitations, and additional experiments are needed to determine the mechanism by which miR-21 promotes fibrosis. miR-21 has a large number of targets that are involved in fibrosis, and identifying these additional targets would shed more light on the development of this process within endometriotic lesions, which could bring more light on the development of the process. We also acknowledge that including other cell lines involved in the process of the development of fibrosis could be very beneficial for the study of the disease. We must consider that the ectopic stromal cell utilized in this study has an origin in an endometrioma, which may be a limitation in understanding the fibrotic pathways in other types of endometriotic lesions.

Our study also had several strengths. We have been able to demonstrate our findings not only in vitro but also in vivo by using our well-established baboon and mouse models to study the role of miR-21 in regulating fibrosis and further compare them with tissues obtained from women with the disease.

## 4. Materials and Methods

### 4.1. Induction of Endometriosis and IL-6 Treatment in the Mouse Model

Animals were maintained in a designated animal care facility according to the Michigan State University’s Institutional Guidelines for the care and use of laboratory animals. All animal procedures were approved by the Institutional Animal Care and Use Committee (IACUC) of Michigan State University. We used the Pgr cre/+ Rosa 26 mT/mG mouse model. This mouse has a double-fluorescent Cre reporter and the ability to express membrane-targeted tandem dimer Tomato (mT) prior to Cre-mediated excision and membrane-targeted green fluorescent protein (mG) after excision (Figure 14A). Six- to eight-week-old mice were injected with estradiol (E2) (0.1 μg/mouse) every 24 h for 3 days, and then endometriosis was surgically induced as previously described [36] (Figure 14B). To access the peritoneal cavity, mice underwent a laparotomy under anesthesia, and a midventral incision (1 cm) was performed to expose the uterus and intestine. The left uterine horn was removed and placed in a petri dish containing sterile PBS. The uterine horn was opened longitudinally and then cut into small fragments. The fragments suspended in 0.5 mL sterile PBS were injected into the peritoneal cavity of the same mouse from which the uterus was taken for autologous implantation, and the abdominal cavity was gently massaged to disperse the tissue. The abdominal incision and wound were closed with sutures, and the skin was closed with surgical wound clips, respectively. After a designated time (15 days, 1 month and 3 months) and during diestrus (Figure 14C), the mice were euthanized, and endometriosis-like lesions were removed and counted using a fluorescence microscope.

For animals treated with IL-6, following induction of endometriosis (Figure 14B), the animals were treated with IL-6 (5 μg/injection) or PBS into the peritoneal cavity for 15 days (Figure 14D). After 15 days, the mice were euthanized, and endometriosis-like lesions were removed and counted using a fluorescence microscope.

### 4.2. Baboon Endometriosis Model

All the experimental procedures were approved by the Institutional Animal Care and Use Committee (IACUC) of the University of Illinois, Chicago, and Michigan State University. Endometriosis was experimentally induced in female baboons (Papio anubis) by intraperitoneal (i.p) inoculation with menstrual tissue on two consecutive cycles, as previously described [58,59,60]. In the cycle before the induction of endometriosis, control eutopic endometrium was obtained at laparotomy on day 10 post-ovulation. Following laparoscopic confirmation of endometriosis at the second inoculation, the animals were evaluated at 3-month intervals post-inoculation and euthanized at 15 months. At necropsy, eutopic and ectopic endometrial tissues were collected, and samples were snap-frozen in liquid nitrogen for RNA/protein extraction or fixed in 10% formalin for morphological and immunohistochemical analysis.

### 4.3. Human Endometrial and Endometriotic Samples

Samples utilized for RNA analysis were obtained with Institutional Review Board (IRB) approval from the School of Medicine of the University of São Paulo. The patients who had a regular menstrual cycle and were primarily being treated for infertility were recruited, as described in previous studies [61,62].

For samples utilized for histology purposes, the study was reviewed and approved by the IRB of Michigan State University and Corewell Health Medical System (Grand Rapids, MI, USA). Written informed consent was obtained from all human subjects. Human endometrial samples were obtained through the Michigan State University’s Center for Women’s Health Research Female Reproductive Tract Biorepository.

### 4.4. RNA Isolation and RT-qPCR

For the human and baboon samples, we utilized all 9 matched eutopic and ectopic tissues from women with endometriosis and 4 matched eutopic and ectopic tissues from baboons with induced endometriosis. Total RNA was isolated using the Trizol reagent (Life Technologies, Carlsbad, CA, USA), and RNA concentration was checked using the NanoDrop 2000 (Thermo Fisher Scientific, Waltham, MA, USA). We performed TaqManTM assay for miR-21 expression analysis and SYBRTM Green assay for *CTGF* using the ViiA7 qPCR System (Applied Biosystems, Foster City, CA, USA). For the microRNA analysis, 100 ng of total RNA was reverse transcribed to cDNA using the TaqManTM MicroRNA Reverse Transcription Kit (4366596, Applied Biosystems, Foster City, CA, USA). RT-qPCR was performed to assess the expression of miR-21 using the TaqManTM Universal Master Mix II with UNG (4440038, Applied Biosystems). The TaqManTM MicroRNA Assays (4427975, Applied Biosystems) which has-miR-21 (000397, Applied Biosystems) and U6 (001973, Applied Biosystems) snRNA were used for microRNA-specific RT-qPCR. For mRNA analysis, 1000 ng of total RNA was reverse transcribed to cDNA using the High-Capacity cDNA Reverse Transcription Kit (4368814, Applied Biosystems). RT-qPCR was performed to assess the expression of the target gene expression using the PowerUpTM SYBRTM Green Master Mix (A25742, Applied Biosystems). The primer sequences for target genes analyzed using RT-qPCR are listed in Appendix A. The expression data were normalized to U6 in the miRNA-specific RT-qPCR and by RPL17 or 18S in the quantitative RT-qPCR. All quantitative reverse transcription-polymerase chain reactions were run for 40 cycles, and the fold change was calculated using the 2^−ΔΔCt^ method [63].

For mice samples, the procedure was the same as with human and baboon samples. Primer sequences for target genes analyzed using RT-qPCR are listed in Appendix A.

### 4.5. In Situ Hybridization (ISH)

For the mouse tissues, we utilized 3 matched eutopic and ectopic tissues with induced endometriosis. Multiplex in situ hybridization assay for the co-detection of miR-21 (FAM2X) and small nuclear 18S (bio2X) was performed as previously described [64]. Four-micron formalin-fixed paraffin-embedded mice tissue was processed for the in situ hybridization assay on a Leica Bond Rx automated stainer. For the baboon samples, we utilized 3 matched eutopic and ectopic from baboons with induced endometriosis. miR-21 staining was calibrated by adjusting probe concentration and fluorescent substrate incorporation time so that no signal was detectable in adjacent normal tissue. Briefly, double-tagged miR-21 (FAM2X) and snRNA U6 (biotin2X) locked nucleic acid–modified DNA probes at 50 nmol/L each were hybridized to tissue slides. Expression of miR-21, U6 and cytokeratin (CK) 19 was assessed with appropriate antibody combinations, followed by sequential rounds of HRP-mediated deposition of appropriate fluorochrome-conjugated tyramine substrates for the baboon samples. Image-Pro Plus software version 7.0 (Media Cybernetics, Rockville, MD, USA) was used for histogram-based image segmentation analysis. miR-21 signal intensity was scored in two locations, glandular epithelium and stroma, on a scale from 0 (no expression) to 3 (high expression). Whole-slide images were acquired using the Aperio Versa imaging system. Pancreatic ductal adenocarcinoma (PDAC) tissues were utilized as quality control. ImageJ Image Analysis software (NIH, v 1.54f) was utilized to determine a digital HSCORE for the quantification of the fluorescence intensity of miR-21.

### 4.6. Masson’s Trichrome Staining

Masson’s trichrome staining was used for the detection of collagen fibers in tissues and to visualize the tissue structure. Paraffin-embedded tissue sections were deparaffinized in xylene and rinsed in absolute alcohol series. After a water rinse, the sections were immersed in Bouin’s Fluid overnight at room temperature. The tissue sections were then stained using the Trichrome One-Step Blue & Red Satin Kit Procedure (#KTTTRBPT StatLab/American Master Tech, McKinney, TX, USA) following the manufacturer’s instructions. The kit steps included the use of Modified Mayer’s Hematoxylin and One Step Trichome Stain followed by the dehydration of the tissue in absolute alcohol series and the final step of xylene before coverslipping the stained uterine tissues and lesions. ImageJ Image Analysis software (NIH, v 1.54f) was utilized to determine the quantification of collagen fibers [65].

### 4.7. Small RNA-Sequencing

Total RNA was extracted from the endometrium and endometriotic lesions from baboons with induced endometriosis for 15 months (*n* = 16) or spontaneous disease (*n* = 8). Small-RNA library preparation and high-throughput sequencing generated an average of 8.9 million reads per sample. Quality and adapter-trimmed reads (Trim Galore v0.3.3) were mapped to human miRNAs from miRbase (release 22) using the miRDeep2 (v0.0.7) pipeline. Differential expression analysis was conducted with edgeR (v3.22.5) using the edgeR-robust method.

### 4.8. Ingenuity Pathway Analysis (IPA)

The miRNAs identified as significantly dysregulated in comparisons between the 15-month uterus samples and lesions were uploaded into QIAGEN Ingenuity Pathway Analysis (QIAGEN IPA). The Ingenuity Pathway Knowledge Base generated a network connecting fibrosis with the dysregulated miRNAs from the 15-month baboon samples.

### 4.9. Cell Culture and Transfection

An immortalized human endometriotic stromal cell line (iEc-ESC) [66] was cultured with phenol red-free DMEM/F-12 (Gibco,Watlhman, MA, USA) medium supplemented with 15% CDS-FBS (Gibco,Watlhman,MA,USA), 100 U/mL of Pen/Strep (Gibco, Watlhman, MA USA) and 1 mM sodium pyruvate (Gibco, Watlhman, MA, USA) at 37 °C. under 5% CO_2_ and 95% air. Following optimization of parameters, Lipofectamine RNAiMAX (Invitrogen, Watlhman, MA, USA) was used to transfect iEc-ESC with 25 pmol of miR-21 mimics (Ambion, Austin, TX, USA) or with 25 pmol of nontargeting negative controls (Ambion, Austin, TX, USA), and RNA and protein were isolated after 24 h. To check the expression of miR-21, RT–qPCR was performed. iEc-ESC cells were also plated at 3 × 10^5^ cells per well in 6-well plates. The following day, the cells were treated with recombinant human IL-6 (206-IL010, R&D Systems, Inc., Minneapolis, MN, USA) (50 ng/mL) in 2% charcoal–dextran-treated fetal bovine serum DMEM/F-12 media. RNA and protein were isolated.

### 4.10. CUT&RUN Assay

The CUT&RUN assay was performed using a kit from Cell Signaling Technology (Cat#86652) and following the manufacturer’s protocol [67]. Briefly, Concanavalin A-coated magnetic beads were treated with Bead Activation Buffer. Ectopic stromal cells (~100,000 cells) were harvested at room temperature and resuspended in a wash buffer containing Spermidine and Protease Inhibitor Cocktail (PIC), and the cell pellet was washed twice. The concanavalin A bead suspension was added and mixed on a tube rotator for 5 min at room temp. Cell/bead conjugate suspensions were resuspended in Antibody Binding Buffer (Digitonin + Spermidine + PIC) containing 5 µL of p-STAT3 antibody (CST cat#9145S) and control reaction with 5 µL of Rabbit IgG (CST, cat# 66362S) and incubated in a tube rotator overnight at 4 °C. The following day, cell/bead suspensions were washed in Digitonin Buffer (+Spermidine+PIC) three times and resuspended in Digitonin Buffer; then, 50 µL of pAG-MNase pre-mix was added to each tube and gently mixed by pipetting up and down, followed by placing the cells on a rotator at room temperature for 1 h followed by two washes in Digitonin Buffer. MNase was activated by adding 3 µL of cold Calcium Chloride to each reaction, and tubes were rotated at 4°C for 30 min. Following this step, the beads were rotated at 37°C for 10 min, followed by centrifugation at 16,000× *g* for 2 min at 4°C, and placed on a magnetic rack until the solution was clear. The enriched chromatin sample in the supernatant was transferred to a fresh 1.5 mL microcentrifuge tube and processed for DNA purification using NucleoSpin Gel and PCR Clean-up kit from MACHEREY-NAGEL GmbH & Co., Germany (cat# 740609.50). The purified DNA was processed for the RT-qPCR analysis using site-specific primer sets (Appendix A), and data obtained were presented as fold enrichment compared to IgG controls.

### 4.11. Statistical Analysis

Data are shown as the mean ± standard deviation (SD). Student’s *t*-test was applied to compare the means of the two groups, and a two-way analysis of variance (ANOVA) with Tukey’s post hoc test for the least significant difference was used for multiple comparisons. *p* < 0.05 was considered statistically significant (2-tailed). GraphPad Prism 9.3.1 (GraphPad Software, San Diego, CA, USA) was used for data analysis.

Differences in miRNA and gene expression between control endometrium and eutopic endometrium and ectopic lesions were compared following normalization against U6 and RPS17, 18S, 36B, respectively. Statistical analyses were performed using Prism 8.3.4 (GraphPad Software Inc. La Jolla, CA, USA). All data were expressed as mean ± SD. Student’s 2-tailed *t*-test was used for comparisons of 2 groups, and 1-way ANOVA with a least significant difference post hoc test was used for multiple comparisons. *p* value less than 0.05 was considered significant.

For mice samples, the fold change expression of miR21 was compared with the uterus. Statistical analyses were performed after log transformation for normalization. A paired *t*-test was performed for the comparison of eutopic versus ectopic. An unpaired *t*-test was performed for uterus versus eutopic tissue comparison. *p* values less than 0.05 were considered significant.

## 5. Conclusions

In this study, we hypothesized that the inflammatory environment, specifically IL-6, in the peritoneal cavity of women with endometriosis could upregulate miR-21 via *STAT3*, leading to an increase in fibrosis in endometriosis (Figure 15). The goals were as follows: determine the mechanisms by which the inflammatory environment driven by IL-6 in endometriosis can upregulate miR-21; determine how miR-21 can modulate the *TGF-β* pathway by the inhibition of *Smad7* to enhance fibrosis; and lastly, understand the interplay between the *TGF-β* pathway/miR21 and *CTGF* and how they contribute towards endometriosis-associated fibrosis.

These studies contribute to a continuously growing body of knowledge of possible factors that regulate fibrosis in endometriosis. We are the first group to show a correlation between miR-21 and its role in the development of fibrosis in endometriotic lesions.

While these studies also discovered new interactions between IL-6 and miR-21, additional studies are needed to fully understand how the inflammatory environment can modulate the action of miRNAs. In addition, our findings align with other recent studies that showed that a persistent activation of *STAT3* via IL-6 signaling is involved in fibrosis in endometriosis [68]. These novel studies have provided new insight into the role of miR-21 and its impact on fibrosis in the context of endometriosis.

## Figures and Tables

**Figure 1 ijms-25-08994-f001:**
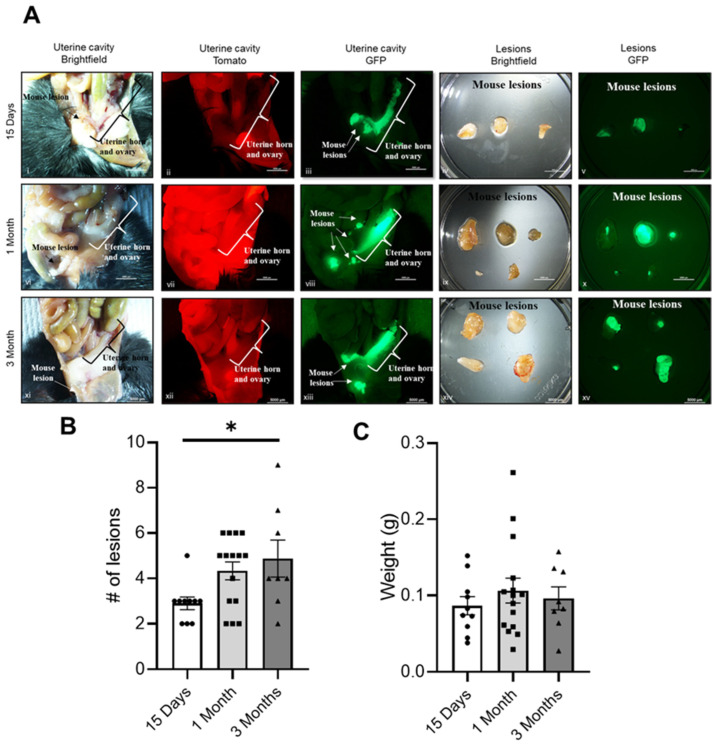
Progression of endometriosis using the Pgr cre/+ Rosa26 mT/mG mouse model: (**A**) Brightfield, Tomato red and GFP images of the uterine cavity of the mouse at 15 days, 1 month and 3 months after induction of endometriosis. Lesions collected from those animals are shown with brightfield and GFP. Scale bar: 5000 μm. (**B**) Comparison of the number of lesions. * *p* < 0.05 and (**C**) weight of the lesions over time. Mice at 15 days (*n* = 10), 1 month (*n* = 15) and 3 months (*n* = 8).

**Figure 2 ijms-25-08994-f002:**
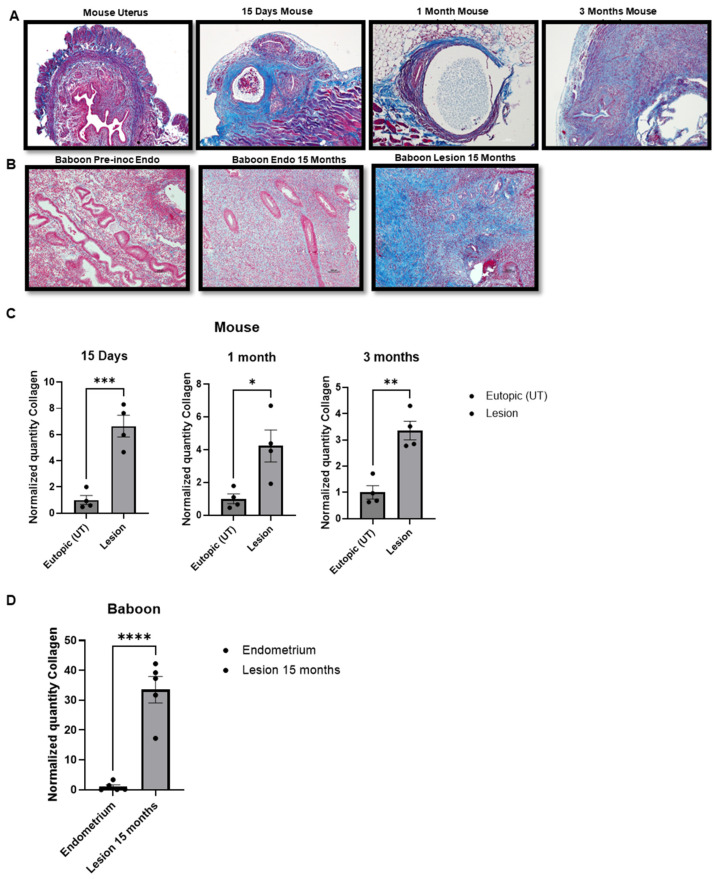
Masson’s trichrome staining: (**A**) Mouse uterus and 15 days (*n* = 4), 1 month (*n* = 4) and 3-month lesions (*n* = 4) Deposition of collagen (blue) around the endometriotic lesion. (**B**) Baboon pre-inoculation endometrium, endometrium after 15 months with endometriosis and 15-month endometriotic lesion (*n* = 5). Deposition of collagen (blue) around the endometriotic lesion. Scale bar: 100 μm. (**C**) Collagen quantification in the mouse model of endometriosis at 15 days, 1 month and 3 months. (**D**) Collagen quantification in the baboon model of endometriosis. * *p* < 0.05, ** *p* < 0.01, *** *p* < 0.001 and **** *p* < 0.0001.

**Figure 3 ijms-25-08994-f003:**
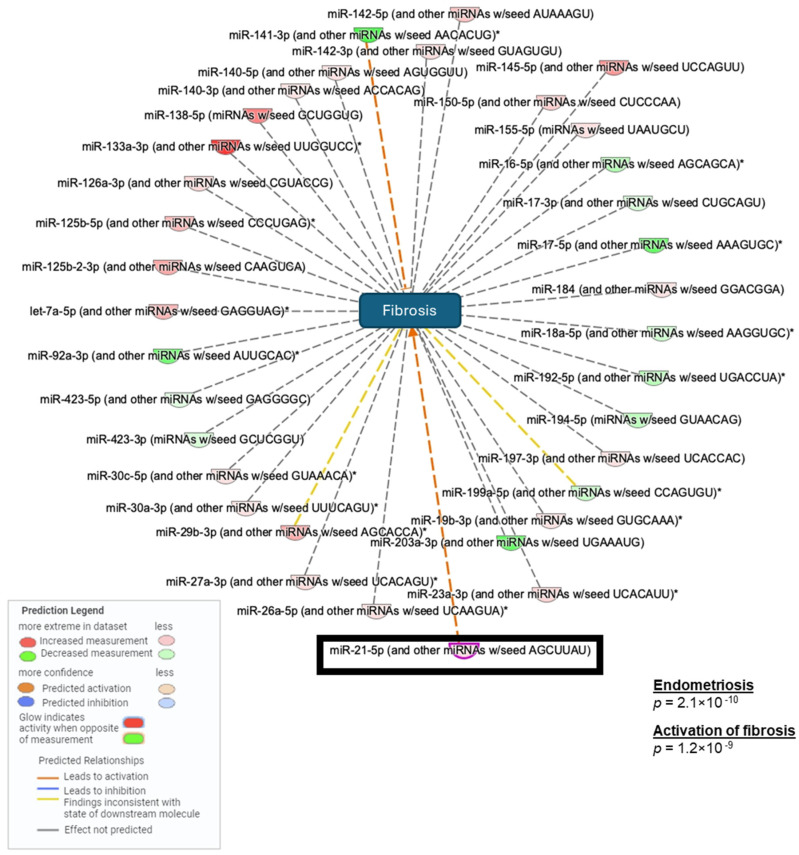
Differentially expressed (DE) miRNAs associated with activation of fibrosis in endometriosis between eutopic and ectopic endometrium in 15-month baboon samples. Small RNA-seq analysis shows the most relevant miRNAs involved in the development of fibrosis, including miR-21. All baboon samples combined, *n* = 24. (*) Multiple DE microRNAs identified within the same familiy of microRNAs.

**Figure 4 ijms-25-08994-f004:**
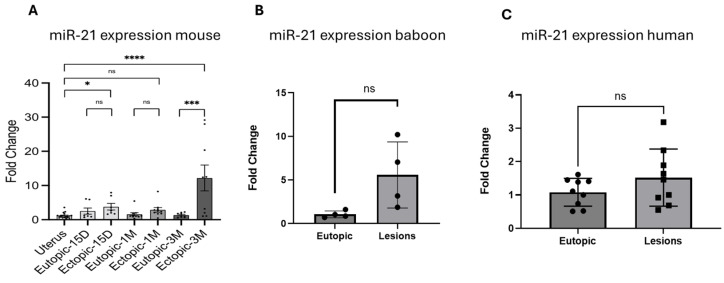
miR-21 expression in mouse, baboon and human lesions: (**A**) mir-21 fold change comparing uterine lesions and matched eutopic lesions in mice at 15 days (*n* = 7), 1 month (*n* = 10) and 3 months (*n* = 9) with endometriosis; (**B**) baboons at 15 months after endometriosis induction (*n* = 4), (**C**) women (*n* = 9) with endometriosis. ns (not significant), * *p* < 0.05, *** *p* < 0.001 and **** *p* < 0.0001.

**Figure 5 ijms-25-08994-f005:**
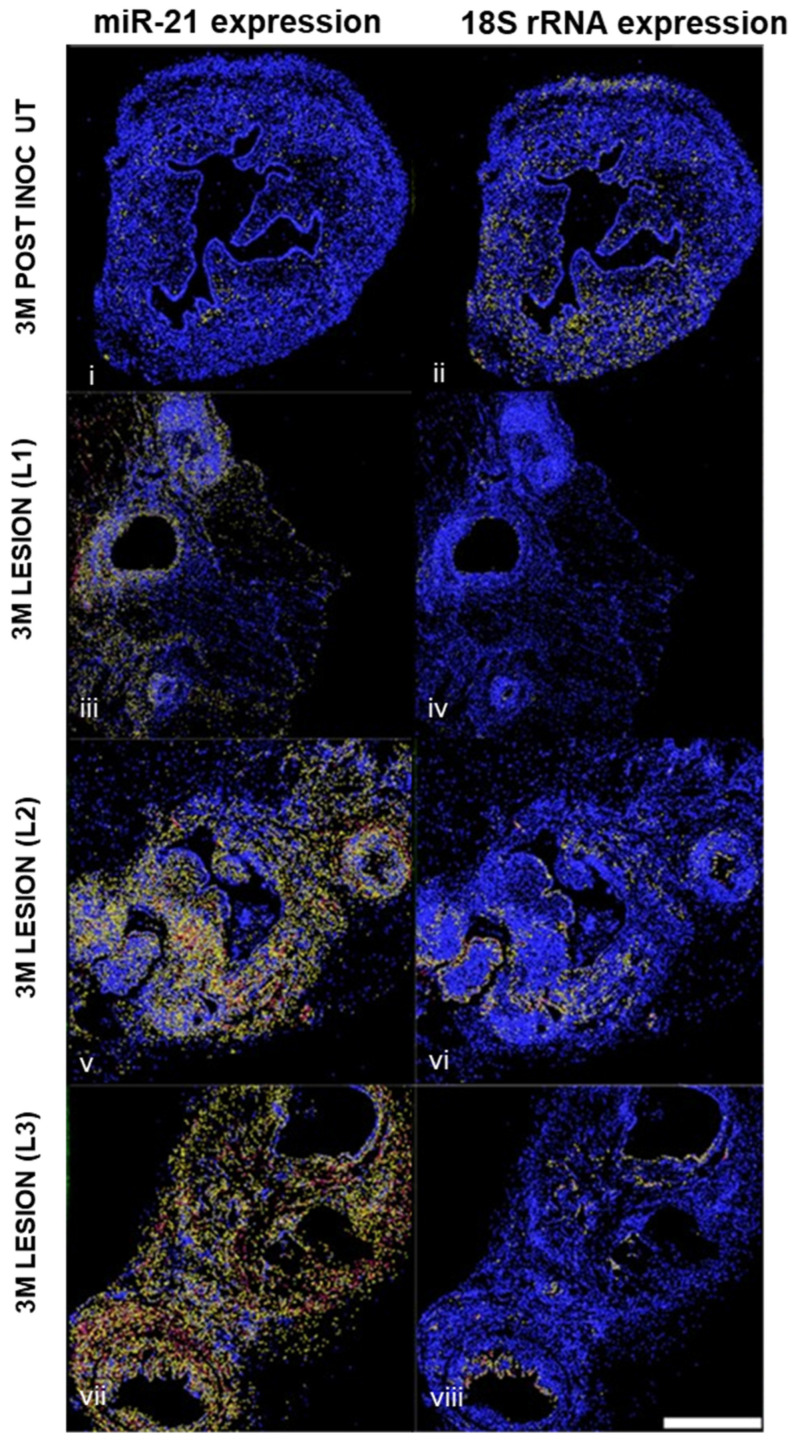
In situ hybridization of miR-21 expression and 18SrRNA expression in mouse uterus and lesions (*n* = 3). Staining intensity was displayed in blue, yellow or red depending on their low to high intensity. Scale bar: 500 μm.

**Figure 6 ijms-25-08994-f006:**
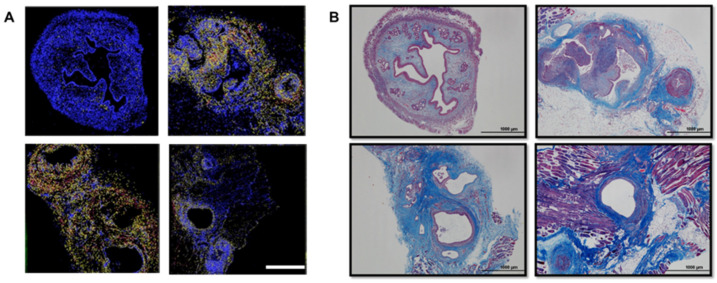
Correlation of miR-21 expression with fibrosis in mouse uterus and three months mouse lesions: (**A**) In situ hybridization of miR-21. miR-21 is present in the stromal cells within the endometriotic lesions; (**B**) Masson’s trichrome staining of the adjacent mouse sections of the uterus and 3-month lesions. Note the deposition of collagen (blue) within the endometriotic lesion. (*n* = 3) Scale bar: 1000 μm.

**Figure 7 ijms-25-08994-f007:**
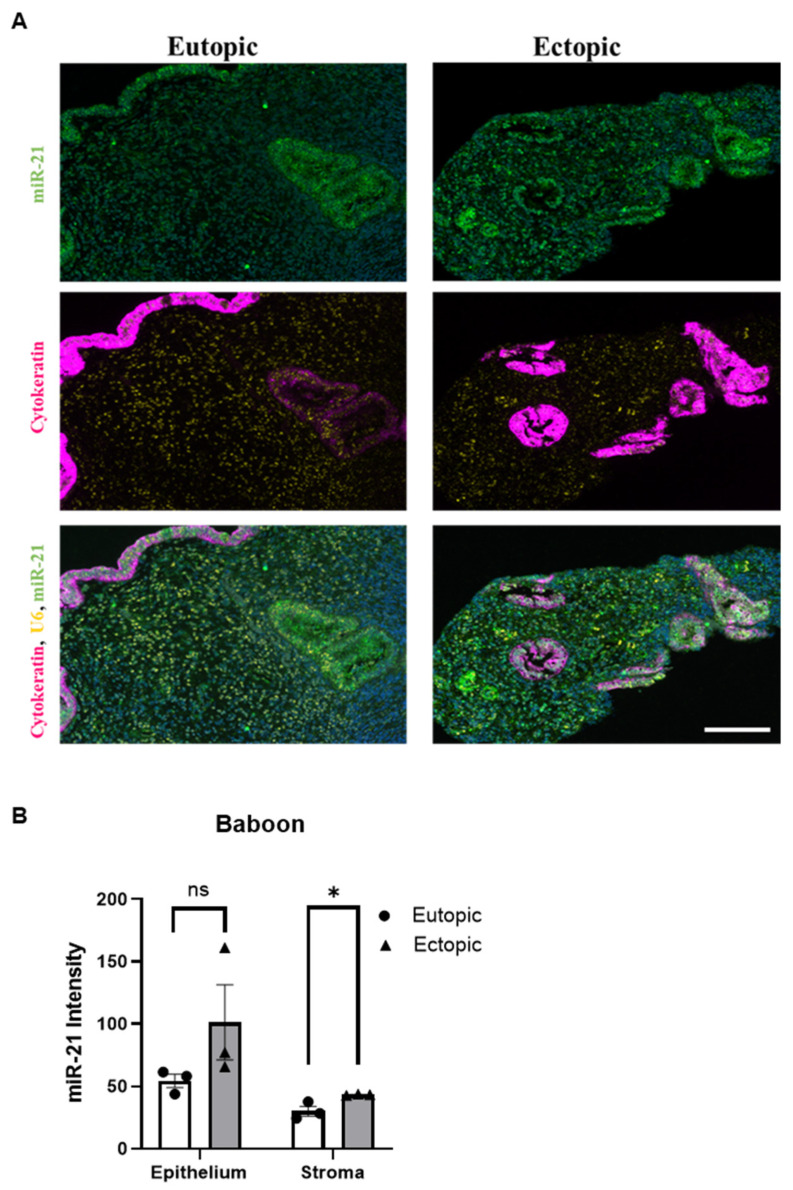
In situ hybridization of miR-21 in the baboon model: (**A**) The left panel shows the in situ hybridization of the eutopic endometrium of the baboon and the right panel shows an ectopic lesion. miR-21 is represented in green, cytokeratin in pink and U6 in yellow. (*n* = 3). Scale bar 300 px. (**B**) miR-21 quantification in eutopic and ectopic . ns (not significant), * *p* < 0.05.

**Figure 8 ijms-25-08994-f008:**
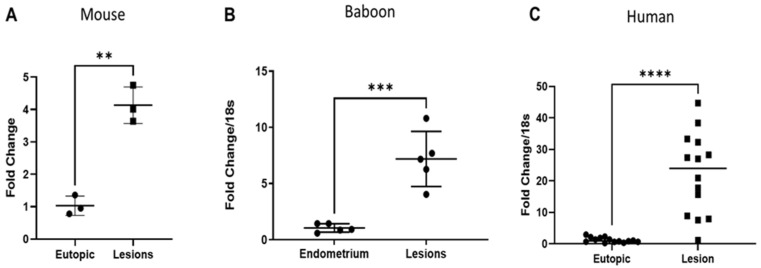
Expression of *CTGF* in (**A**) mouse, (**B**) baboon and (**C**) women with endometriosis. A significant increase in *CTGF* expression in lesions was evident in mouse (*n* = 3), baboon (*n* = 5) and women with endometriosis (*n* = 14). ** *p* < 0.01, *** *p* < 0.001 and **** *p* < 0.0001.

**Figure 9 ijms-25-08994-f009:**
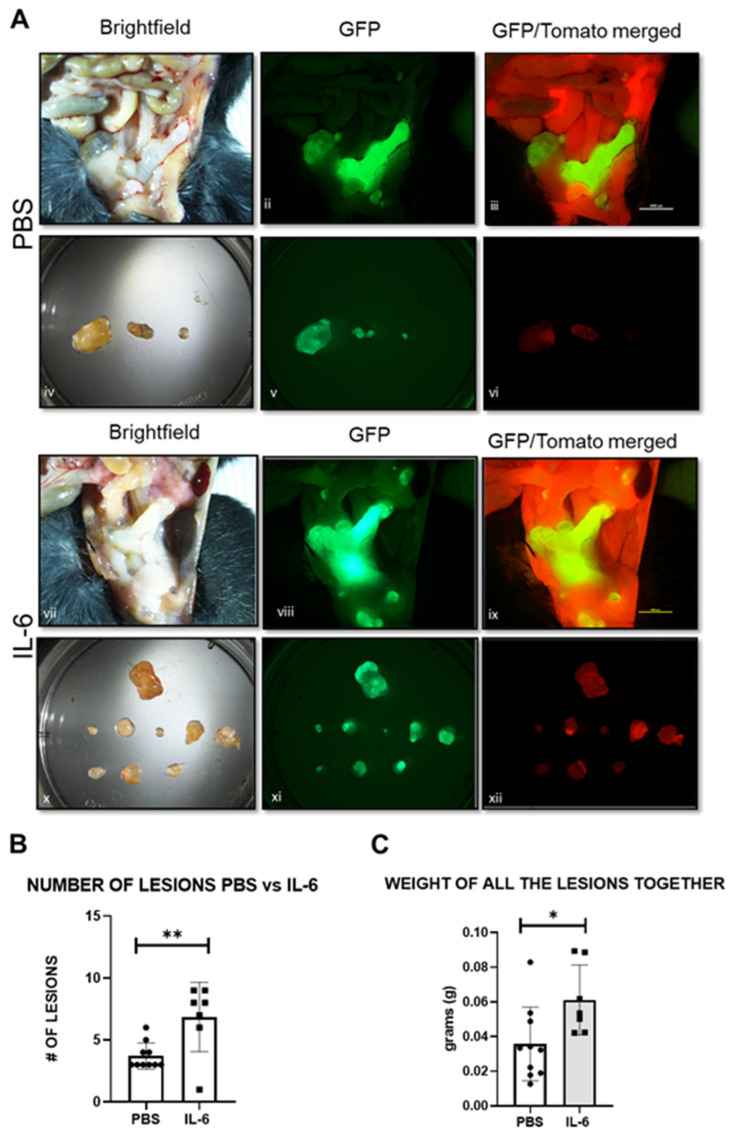
Mouse lesions and uterus following PBS and IL-6 treatment: (**A**) Representative pictures in brightfield (i,iv,vii,x) and fluoresce GFP (ii,v,vii,xi) and GFP/Tomato merged (iii,vi,ix,xii). (**B**) Number of lesions in mice injected with PBS (*n* = 10) or IL-6 (*n* = 7). (**C**) Weight of all the lesions pooled together in animals injected with PBS or IL-6. * *p* < 0.05, ** *p* < 0.01.

**Figure 10 ijms-25-08994-f010:**
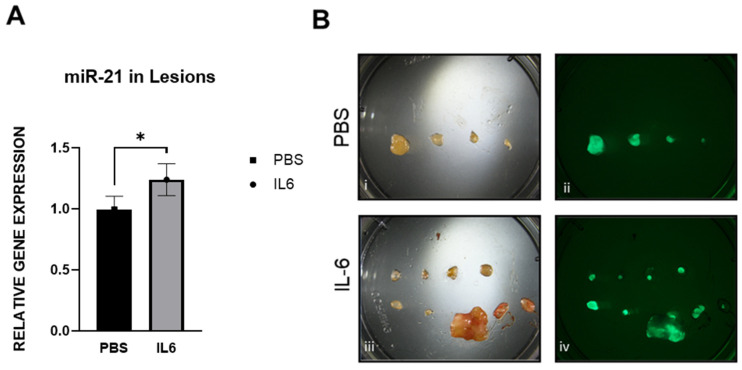
Comparison of miR-21 expression between mice injected with PBS and IL-6: (**A**) Expression of miR-21 in PBS (*n* = 4) compared with IL-6 treated mice (*n* = 3) (**B**) Representative pictures of lesions from PBS and IL-6 mice in brightfield (i,iii) and under fluorescence GFP (ii,iv). * *p* <0.05.

**Figure 11 ijms-25-08994-f011:**
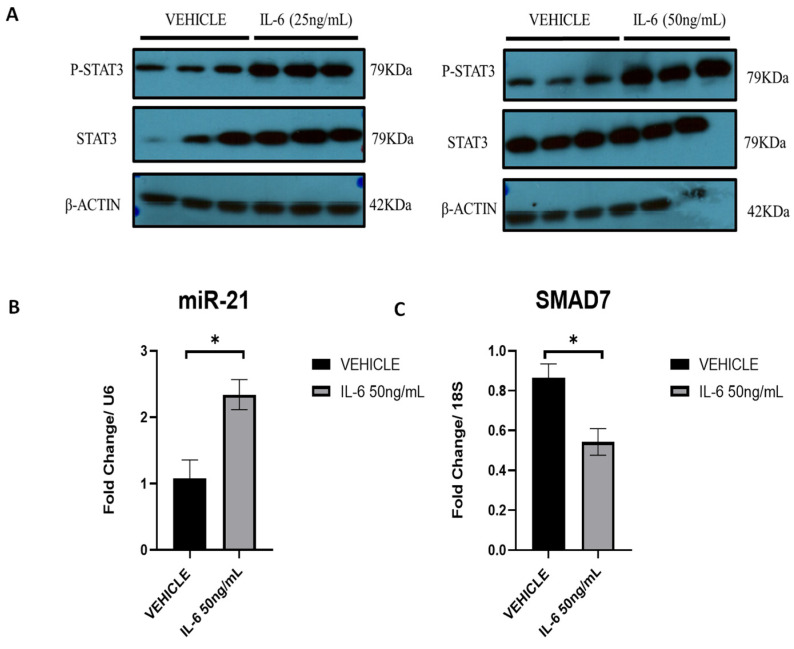
Regulation of miR-21 expression via P-STAT3: (**A**) Western Blot at 24 h of p-STAT3 and STAT3 in ectopic stromal cells in the presence of recombinant IL-6 at 25 ng/mL and 50 ng/mL. (**B**) RT-qPCR analysis at 12 h of the miR-21 expression. (**C**) RT-qPCR analysis of the miR-21 target, SMAD7 expression (*n* = 3). * *p* < 0.05.

**Figure 12 ijms-25-08994-f012:**
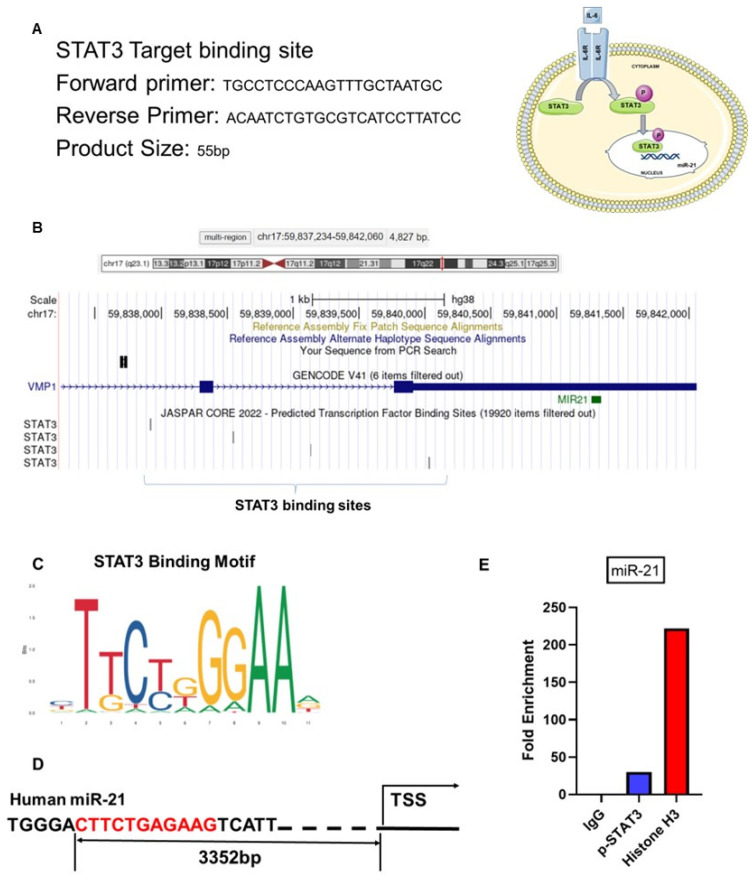
Predicted Transcription Factor binding sites for *STAT3*: (**A**) The primer used for the binding site. (**B**) The results from the transcription factor binding profile database JASPAR CORE 2022. The present study suggests that the expression of miR-21 is increased in ectopic stromal cells via *p-STAT3* binding. (**C**,**D**) The predicted *STAT3* binding site of human miR-21 promoter. (**E**) The binding efficiency of *STAT3* on the human miR-21 promoter enhanced by IL-6 stimulation.

**Figure 13 ijms-25-08994-f013:**
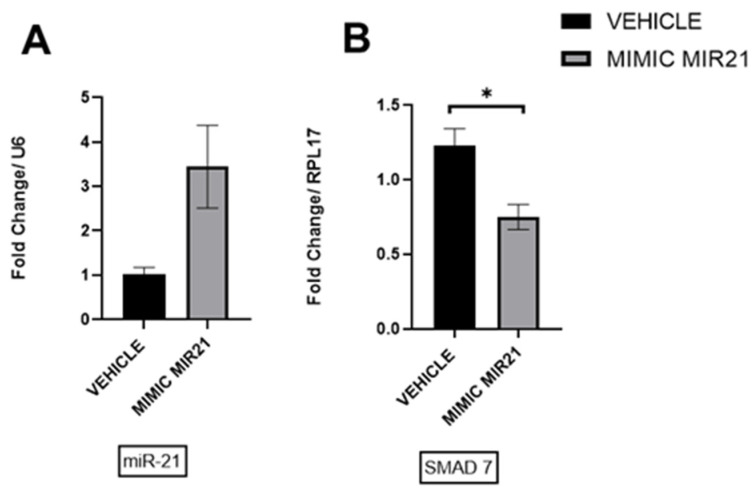
Overexpression of miR-21 in ectopic stromal cells: (**A**) Upregulation of miR-21 (**B**) was associated with the downregulation of Smad7 mRNA (*n* = 3). * *p* < 0.05.

**Figure 14 ijms-25-08994-f014:**
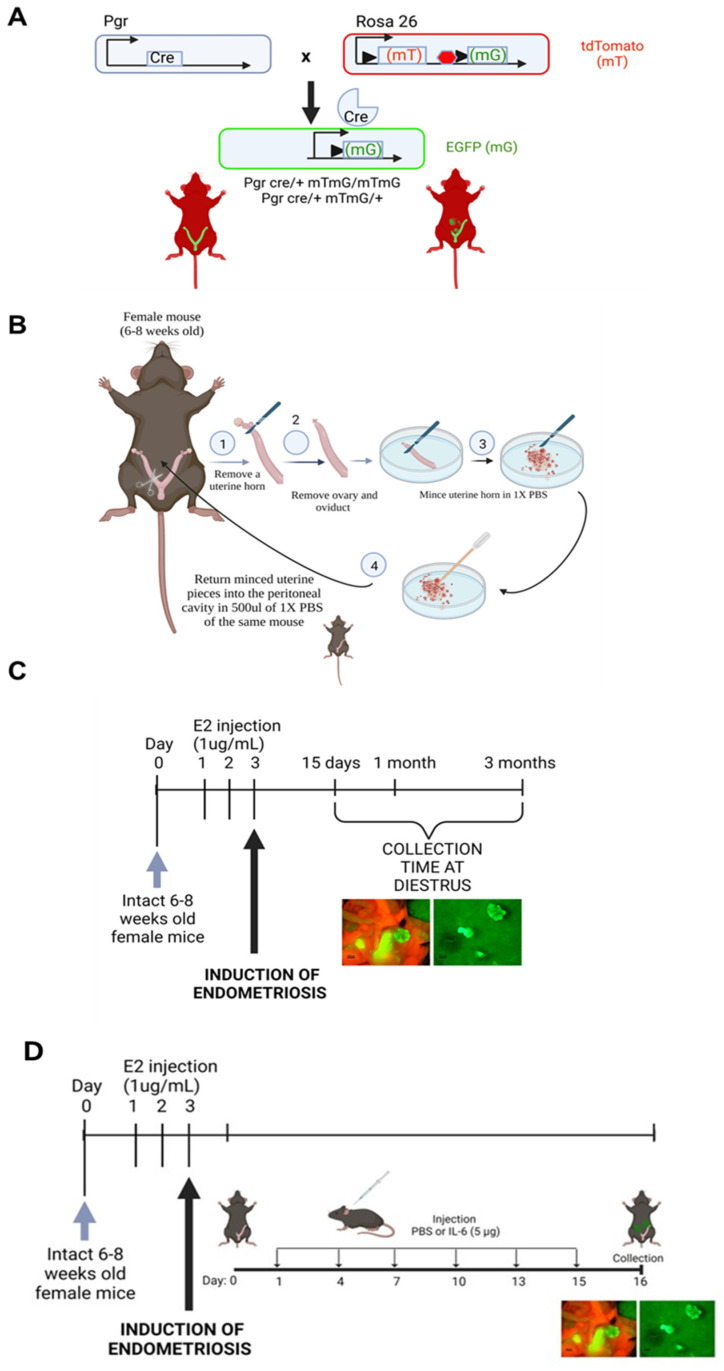
Mouse model used for the induction of endometriosis: (**A**) Schematic diagram of the mT/mG mouse construct before and after Cre-mediated recombination. (**B**) Diagram illustrating the surgical procedure for the induction of endometriosis using the double-fluorescent Cre reporter mouse. (**C**) Schematic diagram of the induction of endometriosis in the mT/mG mouse model of endometriosis. Lesions for the analysis were collected at 15 days, 1 month and 3 months post-induction during diestrus. (**D**) Experimental design of the mouse model of endometriosis and the days of injection of IL-6 or PBS after endometriosis induction. Created with BioRender.com.

**Figure 15 ijms-25-08994-f015:**
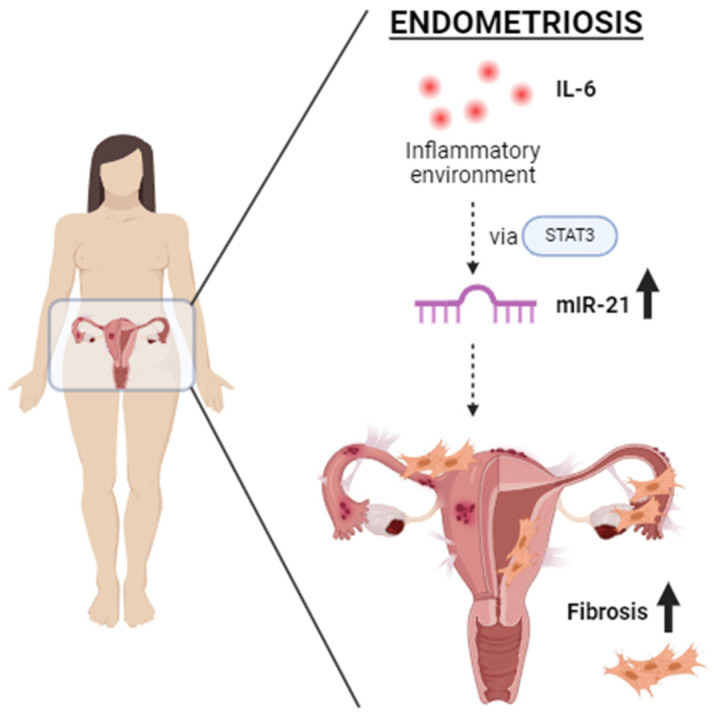
Proposed hypothesis: The inflammatory environment, particularly IL-6, in the peritoneal cavity of women with endometriosis can upregulate miR-21 via STAT3, leading to an increase in fibrosis in endometriotic lesions.

## Data Availability

Raw fastq files were deposited in the National Center for Biotechnology information Gene Expression Omnibus (GSE274922).

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
