# Peer review of "The Regulation of MicroRNA-21 by Interleukin-6 and Its Role in the Development of Fibrosis in Endometriotic Lesions"

_ijms, 2024, doi:10.3390/ijms25168994_

Round 1

Reviewer 1 Report

Comments and Suggestions for Authors

The study by Bernal and colleagues investigates miR-21 in endometriosis and fibrosis. The study is well written (but needs to be checked for punctuation/typos) and adds significant value to the field. On the whole, experiments are robust and several models have been employed to validate their findings. I have several queries/suggestions:

Line 70 – what is meant by ‘the key regulatory pathways’? Please expand.

Line 75 – define ‘CTGF’.

Lines 73 and 77 are repetitive.

Line 104 – there does not appear to be any quantification of Masson’s trichrome staining in Figure 2. The authors go on to claim that fibrosis is increased in the eutopic endometrium by 15 months in the baboon model. This claim is not supported by the data presented, and images of the baboon endometrium (Figure 2B) appear to be from different cycle phases.

Figure 4A – the first bar is labelled ‘uterus’ – how is this different from the eutopic endometrium samples?

Figures 5, 6 and 7 – the expression of miR-21 in the mouse and baboon uterus appears quite different – in the baboon, staining appears most intense in the luminal and glandular epithelium, which is not the case in the mouse. Can the authors comment on these differences?

Figure 6 – the authors claim a correlation between collagen deposition and miR-21 expression. Given the extent of collagen deposition and fibrosis in the lesions, one would not expect to see large numbers of cells in these areas. Indeed, this is supported by the lack of DAPI staining in the fibrotic regions. Where then in miR-21 being expressed in these areas? Is it extracellular?

Line 197 – I disagree that miR-21 is predominantly seen in the stromal cells of the lesions. Visually, the staining in the epithelial cells looks denser and more intense. The authors should perform quantitative analysis of miR-21 expression in the eutopic and ectopic endometrium of the baboon model to make claims on localisation.

Figure 9A – GFP and GFP/Tomato merge do not appear to be the same images overlaid. Further, the GFP/Tomato merge panel appears to show two uterine horns, which should not be the case if one was removed to induce endometriosis.

Figure 11 – all panels of this figure need to make clear the timepoint following IL-6 treatment.

Line 581 – how did the authors determine the appropriateness of parametric statistical tests? Given that some sample numbers were very small (n = 3), non-parametric tests are appropriate.

Comments on the Quality of English Language

Writing is good overall, but would benefit from proof reading to correct typos etc. 

Author Response

REVIEWER’S 1 RESPONSE

Dear reviewer,

Thank you so much for your helpful comments and suggestions. Our response to your concerns are provided below.

-Point 1: Line 70 – what is meant by ‘the key regulatory pathways’? Please expand.

Response 1: Key regulatory pathways refer to regulatory pathways such as tumor development and bone remodeling that we identified in our analysis. The reason we did not further elaborate on these pathways in the paragraph is because we wanted to specifically emphasize the pathways that are commonly associated with endometriosis. To avoid any confusion, we have decided to remove this statement.

-Point 2: Line 75 – define ‘CTGF’.

Response 2: The definition of CTGF - Connective Tissue Growth Factor has been added to the text in line 75.

-Point 3: Lines 73 and 77 are repetitive.

Response 3: Line 77 has been removed from the text.

-Point 4: Line 104 – there does not appear to be any quantification of Masson’s trichrome staining in Figure 2. The authors go on to claim that fibrosis is increased in the eutopic endometrium by 15 months in the baboon model. This claim is not supported by the data presented, and images of the baboon endometrium (Figure 2B) appear to be from different cycle phases.

Response 4:

  1. Quantification of Masson’s Trichrome staining has been added in the figure.
  2. In Figure 2B, the baboon tissues were collected at mid-secretory phase (Pre-inoculation and at 15 months of disease). The reason why the images may look different may be due to the fact that pre-inoculation tissue is collected as an endometrial biopsy at laparotomy as opposed to the 15-month endometrium which is collected at hysterectomy following the termination of the study. This does alter the orientation of the tissue during sectioning of the paraffin blocks.

-Point 5: Figure 4A – the first bar is labelled ‘uterus’ – how is this different from the eutopic endometrium samples?

Response 5: Uterus refers to pre-inoculation Control Uterus. Eutopic refers to post-inoculation Uterus. This information has been added in line 146.

-Point 6: Figures 5, 6 and 7 – the expression of miR-21 in the mouse and baboon uterus appears quite different – in the baboon, staining appears most intense in the luminal and glandular epithelium, which is not the case in the mouse. Can the authors comment on these differences?

Response 6: The primary objective of these figures are to demonstrate that the expression of miR-21 increases  over time in endometriotic lesions compared with the matching uterus in  each model. The difference in the miR-21 expression patterns between mouse and baboon may be due to species differences or/and the differences in collection time points (short in the mouse and long in the baboon).  Figures 5 and 6 (mouse) were collected at 3 months versus Figure 7 (baboon) which was collected at 15 months.

-Point 7: Figure 6 – the authors claim a correlation between collagen deposition and miR-21 expression. Given the extent of collagen deposition and fibrosis in the lesions, one would not expect to see large numbers of cells in these areas. Indeed, this is supported by the lack of DAPI staining in the fibrotic regions. Where then in miR-21 being expressed in these areas? Is it extracellular?

Response 7: The primary point of this figure is to show that the expression of miR-21 correlates with the secretion of extracellular matrix (ECM) by the cells. We understand that fibrosis is a progressive process and expect that the  higher the mir-21 expression is, the higher the secretion ECM from the cells.

-Point 8: Line 197 – I disagree that miR-21 is predominantly seen in the stromal cells of the lesions. Visually, the staining in the epithelial cells looks denser and more intense. The authors should perform quantitative analysis of miR-21 expression in the eutopic and ectopic endometrium of the baboon model to make claims on localization.

Response 8: A quantitative analysis has been added in the figure and a clearer explanation has been provided.

-Point 9: Figure 9A – GFP and GFP/Tomato merge do not appear to be the same images overlaid. Further, the GFP/Tomato merge panels appear to show two uterine horns, which should not be the case if one was removed to induce endometriosis.

Response 9: Thank you so much for your observation. It was an oversight on our part. The image has been replaced with the correct one from the same animal.

-Point 10: Figure 11 – all panels of this figure need to make clear the timepoint following IL-6 treatment.

Response 10:  The correct time points shown for RNA and protein has been added to the figure legend.

-Point 11: Line 581 – how did the authors determine the appropriateness of parametric statistical tests? Given that some sample numbers were very small (n = 3), non-parametric tests are appropriate.

Response 11: We determined the parametric tests versus non-parametric tests depending on the variance analysis of the results.

-Point 12: Proof reading to correct typos.

Response 12: The manuscript has been carefully proofread and we hope the reviewer is satisfied with the improvement in the narrative.

Reviewer 2 Report

Comments and Suggestions for Authors

In this study, the authors aimed to under-19 stand the role of miR-21 and the mechanisms that can contribute to the development of fibrosis by determining how IL-6 regulates miR-21 expression and how this miRNA regulates the TGF-β signaling pathway to promote fibrosis. The study was well thought out but some major points must be addressed. I suggest a new evaluation after the corrections.

Introduction

1)     Line 78-79: The role of miR-21 in fibrosis in endometriosis was investigated in a study (PMID: 38155071; DOI: 10.1016/j.humimm.2023.110746). Please consider this information.

Results

2)     Line 123, 125,127, 129, 130, 131, 133, and beyond: Please, do not use abbreviations for differentially expressed (DE). Using too many abbreviations makes it difficult to follow the paper.

3)     Line 130: "Among the top differentially expressed miRNAs related to fibrosis, miR-21 was identified." What was the basis for relating the miRNAs to fibrosis? Was it based on data from databases or research papers? I understand that your experiment showed differentially expressed miRNAs, but how were they related to fibrosis?

4)     Line 144, 169, 200: Please, write miR-21 and not Mir-21.

5)     Points 2.9 and 2.10 are complementary and hard to follow in the current presentation. Please combine the results.

6)     Lines 255-258: Where are the results (increase number of lesions and increase of the total weight of lesions)?

Material and methods

7)     Line 431: How were the animals euthanized?

Discussion

8)     Line 311-312: The role of miR-21 in fibrosis in endometriosis was investigated in a study (PMID: 38155071; DOI: 10.1016/j.humimm.2023.110746). Please consider this information.

9)     Line 319: Please, write miR-21 and not Mir-21.

10)  Line 374: Please, add this reference PMID: 34103260; DOI: 10.1016/j.rbmo.2021.04.012

Conclusion

11)  Line 612-613: The role of miR-21 in fibrosis in endometriosis was investigated in a study (PMID: 38155071; DOI: 10.1016/j.humimm.2023.110746). Please consider this information.

Figures

12)  Figure 1: Please, change the color of the written part in the uterine cavity brightfield pictures. Sometimes white is better than black. Also, include the statistical test used in B and C.

13)  Figure 3: What means “Endometriosis p= 2.1 x 10-10 and Activation of fibrosis p= 1.2x10-9?

14)  Figure 4: Please, write miR-21 and not Mir-21. Also, add the statistical test used.

15)  Figure 6: It is not possible to determine whether it is miR-21 or 18S, or if it refers to the uterus or 3-month lesions. Please label A and B.

16)  Figures 8, 9, 10, 11, 13: Please, add the statistical test used.

17)  Figure 10: If you compared miR-21 expression between mice injected with PBS and IL-6, why in graphic (A), is PBS expression around 1.5? How long was the treatment?

18)  Figure 11: I would like to see the original blots.

Author Response

REVIEWER’S 2 RESPONSE

Dear reviewer,

Thank you so much for your helpful comments and suggestions. Our detailed responses are provided below.

Introduction

-Point 1) Line 78-79: The role of miR-21 in fibrosis in endometriosis was investigated in a study (PMID: 38155071; DOI:10.1016/j.humimm.2023.110746). Please consider this information.

Response 1: This paper has now been included in the manuscript as a reference.

Results

-Point 2) Line 123, 125,127, 129, 130, 131, 133, and beyond: Please, do not use abbreviations for differentially expressed (DE). Using too many abbreviations makes it difficult to follow the paper.

Response 2: In this paragraph some of the abbreviations have been removed to make the narrative easier to read. Based on the journal guidelines, we have to be consistent with abbreviations that we decided to use and have continued to use them.

-Point 3) Line 130: "Among the top differentially expressed miRNAs related to fibrosis, miR-21 was identified." What was the basis for relating the miRNAs to fibrosis? Was it based on data from databases or research papers? I understand that your experiment showed differentially expressed miRNAs, but how were they related to fibrosis?

Response 3: miRNAs related to fibrosis were based on the results obtained using Ingenuity Pathway Analysis, findings from previous work in our laboratory and previous published manuscripts.  Please check the reference section for more information regarding the papers cited related to this.

-Point 4) Line 144, 169, 200: Please, write miR-21 and not Mir-21.

Response 4: Mir-21 has been corrected and replaced by miR-21 throughout the manuscript.

-Point 5) Points 2.9 and 2.10 are complementary and hard to follow in the current presentation. Please combine the results.

Response 5: We acknowledge that these 2 sections are complementary but point 2.9 describes the in vivo model and the experiments that were performed in the mouse model versus point 2.10, which describes in vitro experiments. Point 2.10 confirms the in vivo data obtained using the mouse model (2.9) which we believe maintains the clarity of the results and confirmation of our in vivo findings in vitro and provides more information related to the mechanisms.

-Point 6) Lines 255-258: Where are the results (increased number of lesions and increase of the total weight of lesions)?

Response 6: The results related to the increased number of lesions and increase of total weight of lesions are provided in Figure 9, referenced in section 2.9 (Lines 229-232).

Material and methods

-Point 7) Line 431: How were the animals euthanized?

Response 7: The animals were euthanized according to the guidelines established by The American Veterinary Medical Association and approved by the IACUC at Michigan State University.

Discussion

-Point 8) Line 311-312: The role of miR-21 in fibrosis in endometriosis was investigated in a study (PMID: 38155071; DOI:10.1016/j.humimm.2023.110746). Please consider this information.

Response 8: This paper has been included in the manuscript.

Point 9) Line 319: Please, write miR-21 and not Mir-21.

Response 9: Mir-21 has been corrected and replaced by miR-21 in the manuscript.

Point 10) Line 374: Please, add this reference PMID: 34103260; DOI:10.1016/j.rbmo.2021.04.012

Response 10: We appreciate the recommendation, but we do not see a clear connection between the results reported in this paper and line 374 in our manuscript.  

(Lines 372-376 in the manuscript): In addition, TGF-β regulates the transcription of several genes involved in fibrosis, including CTGF [46] and studies have suggested that CTGF acts as a mediator of TGF-β-induced fibrotic pathways via miR-21 regulation [22]. Our data supports a potential connection between miR-21 and CTGF during the development of fibrosis in endometriosis.

The paper suggested for inclusion (Functional changes of immune cells: signal of immune tolerance of the ectopic lesions in endometriosis?) describes the potential role of the immune cells and cytokines in peritoneal fluid of women of endometriosis and the role of TH cytokines in endometriosis.

Conclusion

-Point 11) Line 612-613: The role of miR-21 in fibrosis in endometriosis was investigated in a study (PMID: 38155071; DOI:10.1016/j.humimm.2023.110746). Please consider this information.

Response 11: The paper has been included in the manuscript.

Figures

-Point 12) Figure 1: Please, change the color of the written part in the uterine cavity brightfield pictures. Sometimes white is better than black. Also, include the statistical test used in B and C.

Response 12: We agree with the comment but in this figure was not possible to be consistent with the color. The reason why some of the description are in white and others in black is due to the background color of each picture. If we consistently used the same color in each panel the letters and illustrations would not be visible.  

-Point 13) Figure 3: What means “Endometriosis p= 2.1 x 10 and Activation of fibrosis p= 1.2x10?

Response 13: This information is provided in Figure 3 and refers to p values obtained using  Ingenuity Pathway Analysis (IPA). Endometriosis refers to the significance of miRNA expression associated with endometriosis (with the p value= 2.1 x 10-10) and Activation of fibrosis refers to the significance values associated with a fibrotic response (with the p value= 1.2 x 10-9).

-Point 14) Figure 4: Please, write miR-21 and not Mir-21. Also, add the statistical test used.

Response 14: Mir-21 has been corrected and replaced by miR-21 throughout the manuscript.

-Point 15) Figure 6: It is not possible to determine whether it is miR-21 or18S, or if it refers to the uterus or 3-month lesions. Please label A and B.

Response 15: The legend in Figure 6 has been corrected.

-Point 16) Figures 8, 9, 10, 11, 13: Please, add the statistical test used.

Response 16: The statistical analysis undertaken for each of the experiments described in the figures are provided in section 4.11 under Statistical Analysis. We do not think the same information needs to be repeated in each figure.

-Point 17) Figure 10: If you compared miR-21 expression between mice injected with PBS and IL-6, why in graphic (A), is PBS expression around 1.5? How long was the treatment?

Response 17:

(A) The PBS treated mice also develop endometriosis since the minced tissue is injected into the peritoneal cavity and as a result  miR-21 is expressed in these animals. However, injection of IL-6 markedly increased the expression of miR-21. For clarity the data has now been shown as a fold increase following treatment.

(B)The duration of the treatment was 15 days (Please check Figure 14 D).

-Point 18) Figure 11: I would like to see the original blots.

Response 18: Original blots are included as an attachment for your evaluation.  

Original Western Blots. (A) P-STAT3, Vehicle and IL-6 treatment (25ng/mL and 50 mL). (B) STAT3, Vehicle and IL-6 treatment (25ng/mL and 50 mL). (C)β-ACTIN, Vehicle and IL-6 treatment (25ng/mL and 50 mL).

Reviewer 3 Report

Comments and Suggestions for Authors

Dear Authors,

here are some amendments to make to improve the quality of your manuscript:

1- Abstract: it is not cleat if you performed a review or an original article. Please describe briefly the methodology of your research

2- I would suggest to talk in the introduction section about early diagnosis of endometriosis as well as of the understanding of disease progression from adolescence (PMID: 38256683). This topic may also be discussed throughout the text in the context of micro-RNA21. See also your reference nr 17.

3- The structure of the article needs a major revision. Indeed, the correct flowchart should be intro, materials and methods section, results, discussion. Please revise accordingly

4- Results section should be a concise yet efficient reporting of the findings of your experiments. Please avoid inserting citations to other articles or discussion throughout this section.

5- Images are high quality. Well done

6- Please include a strengths and limitations paragraph.

Comments on the Quality of English Language

Minor editing of English language required.

Author Response

REVIEWER’S 3 RESPONSE

Dear reviewer,

Thank you so much for your helpful comments and suggestions. Our responses are provided below.

-Point 1: Abstract: it is not clear if you performed a review or an original article. Please describe briefly the methodology of your research

Response 1: A brief description of the methodology has been added. (Abstract: 200 words limitation)

-Point 2: I would suggest to talk in the introduction section about early diagnosis of endometriosis as well as of the understanding of disease progression from adolescence (PMID: 38256683). This topic may also be discussed throughout the text in the context of micro-RNA21. See also your reference nr 17.

Response 2:  The progression of the disease from adolescence onwards has been included in the introduction of the manuscript. Two new papers (including PMID: 38256683) related to this topic have been added to the references cited in the manuscript.

-Point 3: The structure of the article needs a major revision. Indeed, the correct flowchart should be intro, materials and methods section, results, discussion. Please revise accordingly.

Response 3: We appreciate the comment, but the structure of the article (Introduction, Results, Discussion, Material and Methods and Conclusions) follows the format mandated by the Journal.

-Point 4: Results section should be a concise yet efficient reporting of the findings of your experiments. Please avoid inserting citations to other articles or discussion throughout this section.

Response 4: We removed citations from the Results section and made it more concise as suggested by the reviewer.  

-Point 5: Images are high quality. Well done

Response 5: Thank you so much! We appreciate it.

-Point 6: Please include a strengths and limitations paragraph.

Response 6: A paragraph to address this has been added in the discussion section.

Round 2

Reviewer 3 Report

Comments and Suggestions for Authors

Congratulations, the article is now fine